# Remodeling of the Lamina Cribrosa: Mechanisms and Potential Therapeutic Approaches for Glaucoma

**DOI:** 10.3390/ijms23158068

**Published:** 2022-07-22

**Authors:** Ryan G. Strickland, Mary Anne Garner, Alecia K. Gross, Christopher A. Girkin

**Affiliations:** 1Department of Neurobiology, University of Alabama at Birmingham, Birmingham, AL 35294, USA; rgstrick@uab.edu (R.G.S.); magarner@uab.edu (M.A.G.); agross@uab.edu (A.K.G.); 2Department of Ophthalmology and Vision Sciences, University of Alabama at Birmingham, Birmingham, AL 35294, USA

**Keywords:** glaucoma, optic nerve head, lamina cribrosa, lamina cribrosa cells, scleral fibroblasts, glial cells, intraocular pressure

## Abstract

Glaucomatous optic neuropathy is the leading cause of irreversible blindness in the world. The chronic disease is characterized by optic nerve degeneration and vision field loss. The reduction of intraocular pressure remains the only proven glaucoma treatment, but it does not prevent further neurodegeneration. There are three major classes of cells in the human optic nerve head (ONH): lamina cribrosa (LC) cells, glial cells, and scleral fibroblasts. These cells provide support for the LC which is essential to maintain healthy retinal ganglion cell (RGC) axons. All these cells demonstrate responses to glaucomatous conditions through extracellular matrix remodeling. Therefore, investigations into alternative therapies that alter the characteristic remodeling response of the ONH to enhance the survival of RGC axons are prevalent. Understanding major remodeling pathways in the ONH may be key to developing targeted therapies that reduce deleterious remodeling.

## 1. Introduction

Glaucomatous optic neuropathy (GON) remains the leading cause of irreversible blindness worldwide, and the prevalence is expected to increase in the coming decades [1,2]. Glaucoma is a progressive optic neuropathy which is characterized, in part, by pronounced reorganization of cells in the lamina cribrosa (LC) and peripapillary sclera (ppScl). The variable loading forces imparted on the LC and ppScl by intraocular pressure (IOP), counterbalanced with cerebrospinal fluid (CSF) pressure, result in a region of high strain (tissue stretch) that impacts all ONH cell types and initiates cellular and extracellular matrix (ECM) remodeling. These cellular responses and subsequent ECM remodeling can negatively impact this milieu through which the projecting retinal ganglion cell (RGC) axons must traverse, and this may account for the increased vulnerability to further glaucomatous injury seen in the aged optic nerve or with increasing glaucoma severity (Figure 1).

While IOP lowering remains the only proven treatment, glaucoma can develop and progress even at normal or low levels of IOP. Thus, increasing interest in understanding potential pathways that modulate the pathologic remodeling in the LC and ppScl as a potential approach to develop novel “non-IOP” lowering treatments is emerging, and an abundance of work investigating the mechanisms that underly the ONH remodeling response has been conducted. The aim of this review is to describe the active responses of three major cell populations thought to be most critical to the remodeling response seen in the glaucomatous ONH: LC cells, glial cells, and scleral fibroblasts and to discuss potential therapeutic pathways. While each cell type serves a different purpose, each of these cell populations utilizes similar pathways to respond to the chemical and physical signals presented. Importantly, these responses appear to be consistent between animal models and human tissue culture models of the disease. While therapeutics aims at altering ECM remodeling are a promising potential treatment for glaucoma, it is not the only mechanism that can be exploited clinically.

## 2. Remodeling Response in Aging and Glaucoma

The ONH contains the LC, which is a thin multilayered, reticular load-bearing connective tissue that allows RGC axons and blood vessels to traverse this region of high strain while being supported by glial cells, LC cells, and a load-bearing collagenous matrix. Its unique structure makes it the “weak point” within the sclero-corneal shell where mechanical strain from changes in internal and external pressures on the globe are focused. The LC inserts and anchors itself into the ppScl which provides substantial support to counteract IOP [3,4,5]. In fact, the ppScl may experience the greatest amount of strain in response to elevated IOP [6,7]. The LC and ppScl also receive counteracting pressure from the post-laminar CSF [8]. Since the LC and overlying peripapillary choroid are perfused via branches of the posterior ciliary arteries that are encased in the ppScl, these vessels are subjected to direct mechanical forces as well. Thus, the classic vascular and mechanical theories of glaucoma pathogenesis are inseparably intertwined with the mechanical behavior of the LC and ppScl directly impacting perfusion and vice versa.

As with any load bearing structure, the amount of deformation (strain) experienced by the LC and ppScl is dependent on the morphology and material properties of these tissues, both of which are altered by both age-related and glaucomatous remodeling. Thus, the mechanocelluar response of the tissues, which is driven by strain, directly modifies the structure and material properties (stiffness) of the LC and ppScl which, in turn, alters the strain driving the remodeling. This dynamic creates a feedforward mechanism that may result in an increasingly pathologic milieu. This mechanism may account for the increased susceptibility to glaucomatous injury seen with aging and with increasing glaucoma severity observed across several prospective glaucoma studies [9,10,11,12,13].

A key pathologic characteristic of the glaucomatous ONH is ECM disorganization [14,15,16]. The generation of new ECM is an important component of the glaucomatous response and reorganization of the existing ECM is vital to understand for the development of new treatments. Animal models and ex vivo testing of human cadaveric tissues has shown that the sclera is known to stiffen with age and glaucoma, driving increased strain to the ONH [17,18]. In the non-human primate model of glaucoma, LC structure is dramatically disturbed, and collagen density is altered differently depending on the collagen subtype [14,19,20].

The molecular mechanisms of ECM reorganization center largely on the transforming growth factor-β (TGF-β) pathway, although other pathways are likely implicated. TGF-β is typically inactivated by latency associated peptide (LAP) and enzymes that cleave LAP can consequently activate TGF-β, allowing it to bind to a TGF-β receptor complex and activate downstream Smads that control transcription (Figure 2) [21]. In turn, this pathway results in the increased production of ECM molecules and proteins [22,23,24]. There are several potential mechanisms for TGF-β activation. Interactions with matrix metalloproteinases (MMPs), integrins, and thrombospondin (TSP) can all trigger TGF-β activation [25,26].

There are a multitude of enzymes in the MMP family with various substrate specificity and all contribute to the degradation of ECM components such as collagen, gelatins, laminin, and more [27,28]. In the glaucomatous ONH, MMP-1, -2, -3, and -14 have all demonstrated upregulation [29]. The TSP family has been implicated in a variety of fibrotic pathologies due to its ability to activate TGF-β, and it has implications in ECM remodeling and in IOP levels in knock-out mouse models [30,31]. However, evidence suggests that expression levels of TSP isoforms may vary depending on disease stage, such as low TSP-4 expression in early glaucoma that increases in late stages of the disease [32]. Lastly, integrin signaling-mediated activation of TGF-β may also be more pertinent to glial cells and vascular endothelial cells in the ONH [33]. In total, the elements of this pathway have multiple different implications on the profibrotic responses of the cell types discussed below. Cellular responses include generation of newly synthesized ECM, ECM editing, cellular migration, and cellular contractility. Overall, there are four cell types involved in ONH remodeling including the lamina cribrosa (LC) cells, which are in contact with the laminar beams, astrocytes and microglia cells within the pores of the LC, and scleral fibroblasts within the ppScl.

## 3. Lamina Cribrosa Cells

LC cells are typically differentiated from regional astrocytes by the lack of GFAP, constitutive expression of α-smooth muscle actin (α-SMA), alternate shape, and localization among other factors [34,35]. While LC cells typically maintain the supportive laminar beams and ECM through production of collagen, elastin, and fibronectin [35], they are also capable of dynamic reactions to external stimuli that can alter the properties of the ECM. For example, human LC cells in culture exposed to mechanical strain demonstrated altered expression levels of multiple genes that implicate ECM components, cell proliferation, growth factors, and cell surface receptors [36]. In addition to mechanical strain, LC cells can also respond to oxidative stress by upregulating fibrotic genes and production of collagen and α-SMA [37]. Human LC cells cultured in hypoxic conditions also demonstrate increased expression of ECM-related factors such as macrophage migration inhibitory factor and discoidin domain receptor [38,39,40]. This evidence shows that mechanical strain, oxidative stress, and hypoxia, all potentially relevant to the pathogenesis of primary open-angle glaucoma (POAG), cause LC cells to express and secrete collagen as well as other fibrotic molecules.

LC cells are likely to play a critical role in reorganization and remodeling of the LC in response to mechanical activation also through the TGF-β pathway. Specifically, MMP-2 expression and activity are both increased in response to glaucomatous conditions [24,34,41]. MMP and TSP1 expression is upregulated in LC cells responding to mechanical stress [24,36]. Increased expression and secretion of active MMPs that digest ECM components are likely a major component of ECM disorganization in response to strain. Additionally, LC cells use the ECM matrix as a scaffolding and with localized degradation, LC cells may use the detachment to migrate within the ONH which could underlie the ability of the LC to migrate posteriorly within the ONH [42,43]. There is evidence that application of human TGF-β2 also stimulates MMP activity in porcine LC cells which further supports the role of this pathway in another animal model [44]. Unfortunately, rodent ONHs do not contain LC cells and, though it has been noted previously, there remains no published attempts to culture primate LC cells [45].

As previously mentioned, these notable ECM modifying proteins can also function as activators of latent TGF-β. However, there is also evidence of the reverse: active TGF-β and the subsequent Smad transcriptional regulation pathway controls the expression level of its own activating partners. For instance, application of TGF-β1 to cultured human LC cells induced greater expression of TSP [23]. This evidence suggests that the initial triggers of the TGF-β activation initiate a feed-forward mechanism of ECM remodeling in the LC that has deleterious effects on the axons of the RGCs in the region [26]. This feed-forward signaling mechanism has been demonstrated in the other cell types described below.

TGF-β, as well as mechanical and oxidative stress, can also influence aspects of ECM regulation in LC cells through calcium-dependent pathways. For instance, LC cells cultured from glaucomatous human eyes demonstrated dysregulation of calcium, such as high intracellular levels in response to previously mentioned stimuli and a reduced ability to sequester free cytoplasmic calcium [46,47]. Increased levels of cellular calcium can have a variety of effects on signaling pathways due to the dynamic nature of calcium as a second messenger. Of those pathways, the activation of nuclear factor of activated T-cells (NFAT) may be most relevant to LC cells. In short, calcineurin can bind calcium to calmodulin, which can dephosphorylate NFAT. NFAT then complexes with transcription factors to influence transcription of genes including those that modulate the ECM [48]. While inhibition of this pathway may aid in the treatment of glaucoma, it is not fully understood what precedes the loss of calcium regulation in glaucomatous LC cells. However, one potential preceding factor may be the presence of transient receptor potential canonical (TRPC) channels, a class of voltage-independent channels that preferentially respond to calcium ions and are not necessarily dependent on stimuli such as TGF-β or oxidative and mechanical stresses [49,50]. Interestingly, isoforms of TRPCs, such as 1 and 6, are significantly overexpressed in glaucomatous LC cells cultured from human ONHs [51]. The increased presence of these channels may disturb homeostatic levels of calcium in LC cells, thereby inducing dysregulation. Additionally, these channels modulate transcription of ECM components such as TGF-β, α-SMA, collagen, and MMPs likely through the NFAT pathway described previously [49,51]. Furthermore, TRPC-1 and -6, at least in cancerous cells of the central nervous system, also regulate cell migration [49,52,53]. While these mechanisms of migration have not directly been shown in LC cells, it is possible that glaucomatous LC cells overexpressing these TRPCs may initiate signaling cascades that increase the degree to which LC cells migrate within the LC, potentially contributing to posterior LC migration.

## 4. Glial Cells

The ONH also contains a resident population of glial cells that create the blood–brain barrier (BBB) and myelinate the RGC axons in the post-laminar region. Dormant microglia are also primed for reactionary responses to local insult or disease [29,54]. Astrocytes, like LC cells, demonstrate mechanosensitive properties and are reactive to glaucomatous conditions in humans and other animal models [55,56,57,58]. In the healthy ONH, astrocytes typically support the BBB, but type 1B astrocytes, the dominant subtype in the ONH, can also assist the LC cells in producing the ECM in response to glaucomatous conditions [15,59,60,61]. Furthermore, astrocytes make connections with other astrocytes and LC cells which could aid in coordinating responses to mechanical strain. Actin reorganization of astrocytes in response to elevated pressure can occur within hours of IOP elevation in rodents and reorganization to baseline may happen over the same time scale, or perhaps even days [62,63]. Reactive astrocytes also produce MMPs which could serve a similar purpose as suggested previously; to sever connections to allow cellular displacement as well as rearrangement of the ECM [29]. As discussed above, MMP function ties in closely with TGF-β, and evidence shows that astrocytes utilize the TGF-β pathway in response to glaucoma as well [22,64,65]. Also prominent to the TGF-β pathway is connective tissue growth factor (CCN2; referred to here as CTGF), a significant binding partner of TGF-β that has been shown to affect the TGF-β pathway, and it is required for Smad1 but not Smad3 activation [66]. CTGF is a mediator of ECM synthesis in the anterior segment as well, as demonstrated by a murine model with increased secretion of CTGF resulting in trabecular meshwork (TM) remodeling and increased IOP [67]. Further evidence in mice shows that the astrocytic levels of CTGF in the ONH increases in glaucomatous animals as a result of elevated IOP and stiffness, which agrees with the observation that there are elevated levels of CTGF in glaucomatous ONHs of humans as well [68]. At least in mice, CTGF seems to be predominantly expressed by astrocytes in the ONH [69], but there is reason to suggest that CTGF may affect other cell types such as LC cells [37].

Integrin signaling in astrocytes may also be involved in the LC cell in detecting tissue strain and inducing cellular migration and reattachment [33]. However, these are not the only molecules involved in astrocytic remodeling as myosin light chain kinase has increased expression in response to mechanical strain of astrocytes and is implicated in cellular migration [70]. Moreover, phosphoinositide 3-kinase, protein kinase C, and tyrosine kinase have also be implicated in migration [71]. Astrocytes also detect mechanical strain with TRPCs which may provide early responses to initial IOP increases. For example, in an induced mouse model, reactive astrocytes respond within one hour of an IOP increase, likely mediated by TRPC isoforms sensitive to stretch [72]. As previously mentioned, this TRPC-NFAT pathway can induce transcriptional changes related to the ECM and can influence cellular migration.

Astrocytic responses are not limited to mechanoreceptors as hypoxia can also trigger responses. Hypoxia-inducible factor-1α (HIF-1α) is a transcriptional factor that is upregulated in response to hypoxic conditions and plays a role in cellular metabolism, proliferation, and angiogenesis [73]. The link between the glaucomatous ONH and HIF-1α was first noted by examining human eye post-mortem, but it has been noted in glaucomatous dogs as well [39,74]. These findings have been replicated in induced rodent models of glaucoma, and the evidence indicates that HIF-1α activation is localized to astrocytes of the retina and ONH [75,76]. There is currently no explanation for why HIF-1α responses are localized to astrocytes and not microglia or RGCs in these models. However, it does indicate that global hypoxic conditions of the ONH, at least on these early timescales, cannot explain RGC dysfunction due to ischemia. Alternatively, PACE4, a subtilisin-like protein convertase, is known to increase expression in response to hypoxia, which may also occur due to vascular compression or primary vascular or vasospastic disease [39,77]. PACE4 also displays constitutive expression and activity in glial cells across the retina, but more so in the ONH [78]. This may be an important factor to consider given the evidence that the PACE family interacts with inhibitors of MMPs, tissue inhibitors of matrix metalloproteinases (TIMPs), as well as TGF-β [79].

Astrocytes at the post-laminar, myelination transition zone (MTZ) are also of interest due to the potential posterior shift of the LC that may be signaled by mechanotransduction of the cells. Specifically, galectin-3 (also known as Lgals3 or Mac-2) has been shown to be upregulated and involved in astrocytic phagocytosis of RGC axons [80]. Additionally, recent evidence in an inducible murine model has shown that astrocytes near the MTZ react by projecting longitudinal processes into the axonal bundles of the ONH, rather than encasing the axons, perhaps contributing to phagocytosis [56]. There is also reason to believe that such phagocytotic absorbance of mitochondria localized to the axons may precede RGC degeneration [81,82,83].

Microglia are also present in the ONH and are activated in glaucomatous eyes [84]. Similar to LC cells, activated microglia express both TGF-β and MMPs which are not produced in the microglia of healthy ONHs [29]. While microglia are likely incapable of the secretion of ECM molecules, there is mounting evidence that suggests that microglia across the central nervous system are highly active in the maintenance of the ECM through reorganization using MMPs, TSPs, and other similar proteins [85,86]. These processes may be important in the formation of glial scarring, a deposit of new ECM that may not be beneficial to the RGC axons [87]. While this produces a physical obstruction within the ONH that can damage axons, deposits of proteoglycans such as tenascin, which is produced by astrocytes in a mechanically independent model of glaucoma, may provide some initial protection to the axons [88,89,90]. Furthermore, tenascin is a substrate on which MMPs can act and remodel. Activation of microglia can be dependent on integrin signaling, detection of damaged cells, and growth factors such as TGF-β [33,86,91,92]. While some components of astrocyte activation are necessary for cellular repair and neurotrophic factors, persistent activation can lead to secretion of cytotoxic molecules that are likely further detrimental to the RGC axons [93].

The reactivity of both astrocytes and microglia contributes to the neuroinflammatory conditions of the ONH that may negatively impact the surrounding milieu. This perspective of GON is complex and has generated a rapidly emerging line of work which has recently been adequately reviewed [94,95,96,97,98].

## 5. Scleral Fibroblasts

The ppScl, or scleral flange, is the portion of sclera immediately surrounding the scleral canal which provides an anchoring point for the LC. It also contains the penetrating branches of the posterior ciliary arteries that perfuse the LC and overlying choroid. The ppScl, as with the remaining sclera, is composed of a dense, collagenous ECM interspersed with resident fibroblasts that maintain the ECM [99]. Similar to the other cell types, the scleral fibroblasts of the ppScl also produce ECM remodeling factors in response to glaucomatous conditions as demonstrated in human tissue and primate, and mouse models [19,100]. A notable characteristic of these cells is that when active and responsive, they differentiate into myofibroblasts consequently expressing α-SMA [101,102,103]. Differentiation of scleral fibroblasts can be caused by a mechanosensitive response to increased pressure and leads to the secretion of ECM materials, such as collagen, and ECM editors, such as MMPs and TIMPs [104,105,106,107]. Myofibroblast differentiation is also partly dependent on Src-kinase pathways as inhibitors of the pathway, such as dasatinib, can restrict the process [108]. Myofibroblast differentiation requires transcriptional changes which can be seen as soon as 30 min after mechanical strain is applied to human tissue culture [109].

Scleral fibroblasts use the collagenous matrix as the point of cell adhesion. This collagenous matrix of the ppScl is morphologically distinct from the rest of the sclera. Specifically, collagen in the ppScl runs in a circumferential pattern around the ONH while collagen in the posterior sclera is arranged in a “basket-woven” pattern [110,111]. Interestingly, this pattern correlates with distribution and alignment of scleral fibroblasts. Fibroblast density increases in proximity to the ONH, and fibroblast projections are highly aligned with collagenous structures [112]. There is also limited evidence to suggest that such fibroblast density gradients may exist in mouse as well [113]. Given this precise alignment of fibroblasts and collagen, these cells likely play a role in the detection of tissue stretch. The reaction of fibroblasts may also differ based on localization as α-SMA expression appears to disrupt fibroblast projection alignment with collagen in the peripheral sclera, but not the ppScl, and fibroblast orientation is most altered when cells detect both strain and TGF-β signaling simultaneously [112]. These synergistic processes likely reinforce chronic glaucomatous remodeling of the ppScl, altering its biomechanical properties.

Properties of scleral fibroblast differentiation and proliferation are partly mediated by both Yes-Associated-Protein (YAP) and Rho-associated protein kinase (ROCK) [114,115]. A notable trait of myofibroblasts is their expression of α-SMA, which may aid in acutely altering scleral stiffness and in cell migration [116]. Furthermore, fibroblast migration is also associated with ROCK and YAP as inhibition of both leads to reduction in rates of migration as well as the contractile abilities associated with α-SMA [114,115]. ROCK inhibitors thus may be a potential treatment that is currently used to increase aqueous outflow in the anterior segment.

Similar to LC cells, TGF-β and Smad-based transcription play a role in fibroblast responses as well. Application of TGF-β to cultured scleral fibroblasts induces higher levels of α-SMA expression and contractility [115] and binding partners of Smad are upregulated in scleral fibroblasts responding to stretch [109]. Additionally, YAP and Smad3 are shown to interact in human scleral fibroblasts which undergo strain [114]. Within an induced mouse model of glaucoma, upregulation in both TSP and integrin expression, both activators of TGF-β, have been demonstrated [104]. Furthermore, an ECM remodeling protein, TGF-β inducible protein (TGF-βip), is known to express in response to TGF-β signaling pathways, especially in collagen rich tissues [117]. There is evidence of the presence and secretion of TGF-βip in human and non-human primate sclera, and TGF-βip has binding properties with integrins at the cell surface in human scleral fibroblasts, which are implicated in stretch detection and in the modulation of the biomechanical properties of the cell [118,119,120]. TGF-βip can also inhibit fibroblast adhesion to collagen, which likely affects remodeling and cellular migration [119]. Taken together, these results suggest that scleral fibroblasts in glaucomatous conditions use similar signaling pathways to LC cells, such as TGF-β, to differentiate, migrate, and induce extensive remodeling of the sclera and ppScl.

## 6. Discussion

Glaucoma treatment is difficult due to its complex, incompletely understood, pathophysiology, and while IOP lowering is impactful, it does not universally prevent the progression or development of the disease. These approaches to lower IOP focus on reducing the mechanical stress applied to the optic nerve head. Altering the material properties or morphology of the LC and ppScl by manipulation of the processes involved in ONH remodeling has the potential to increase the resilience of the ONH to the stress induced by changing IOP and promote RGC survival. However, it remains unclear what mechanical properties of the sclera and LC are beneficial and what properties are harmful. For example, there have been hypotheses that increased stiffness could resist elevated IOP levels [17]. However, a stiffer scleral may increase the strain experienced by the OHN by directing stress to the weakest point in the eye wall. Multiple scleral stiffening compounds such as glyceraldehyde, glutaraldehyde, and genipin have failed to demonstrate RGC protection in rodent models or tree shrews thus far [121,122,123] and may be harmful to the retina as well [124]. Alternatively, perhaps reducing scleral stiffness could alleviate certain cases of glaucoma through the application of collagenase or other compounds that break down glycosaminoglycans [125,126,127]. Although the evidence is limited, one study showed that rats with experimentally induced glaucoma and subsequently treated with a glycosaminoglycan digesting agent via intravitreal injection demonstrated preservation of RGC dendritic fields [128]. While it is promising that the manipulation of scleral material properties may impact glaucoma development, it is unclear how a collagenous LC may respond to approaches that increase or decrease LC stiffness in terms of RGC survival.

Several therapeutic approaches targeting the cells that create the ECM have been suggested in an attempt to inhibit molecular pathways that trigger ECM secretion (Table 1). Notably, many of the drugs under investigation target some element of the TGF-β pathways referenced previously. For instance, losartan inhibits the G-protein-coupled receptor (GPCR) angiotensin 1 and is a target for the TGF-β ligand [129]. When losartan was administered orally to mice with experimentally induced glaucoma, RGC loss was prevented, likely by inhibiting the degree to which scleral fibroblasts could remodel the ECM [130]. However, a side effect of losartan is decreased blood pressure which could reduce ocular perfusion pressure, a critical risk factor for glaucoma.

Additionally, the competitive antagonist LSKL, which can cross the BBB, may be another candidate for the treatment of glaucoma. LSKL prevents TSP1 from binding to LAP which prevents activation of TGF-β and its downstream pathway. The result is that LSKL administration to rodents leads to reduced fibrosis with minimal side effects [26,256,257]. Despite the potential, there has not been any published evidence that LSKL can ameliorate the glaucomatous ONH and preserve RGCs [30]. CTGF, a primary interactor of TGF-β, is also profibrotic, and inhibition of CTGF in cultured human LC cells using the monoclonal antibody FG-3019 blocked ECM synthesis in these cells [37]. The antibody FG-3019, or pamrevlumab, is currently being evaluated in clinical trials for idiopathic pulmonary fibrosis, but its effectiveness beyond cultured cells in glaucoma has not yet been illustrated [258]. While intriguing, there are also a multitude of other potential therapies or pathway targets of TGF-β that could be beneficial for the treatment of glaucoma [259]. Future therapeutic pathways may further probe the relationship between TGF-β and both bone morphogenic proteins (BMPs) and Wnt signaling; two candidates that may inhibit profibrotic responses [260,261].

Another method to prevent remodeling in the ONH is to prevent the synthesis of collagen by LC cells and astrocytes before it is secreted. The drug tranilast is known to inhibit collagen synthesis and has been shown to work on cultured human LC cells and astrocytes [228]. The inhibition of rho-associated protein kinase (ROCK) has also been shown to prevent scleral fibroblast from exhibiting myofibroblast features, limiting contractility and expression of α-SMA [115]. In fact, ROCK inhibitors, such as K-115 or ripasudil, are currently being investigated in clinical trials for glaucoma treatment [222,262,263,264].

Some common currently available systemic and topical treatment may also potentially impact glaucoma development. Statins have been shown to prevent TGF-β-mediated activation of MMPs through ROCK pathway inhibition in cultured human eye [65]. Additionally, simvastatin has demonstrated a protective effect on RGC survival in a mouse model of retinal ischemia via elevated IOP [240]. Whether or not statins could be effective at treating human cases of glaucoma with little side effects is unknown [265], but administrative database studies and meta-analyses have suggested a potential protective effect, slowing the development of glaucoma [266,267,268]. Lastly, prostaglandin analogues, the most common IOP-lowering treatment for glaucoma, work by upregulation of MMPs via the prostaglandin F receptor. These same pathways that alter the ECM of the iris and ciliary body to lower IOP are also involved in glaucomatous remodeling in the posterior pole [269]. While it is unknown whether these compounds would reach therapeutic levels in the posterior pole with topical administration, methods of delivery could be developed to impact the ONH more directly.

In summary, there is intriguing emerging evidence that manipulation of the mechanical properties of the sclera and/or ONH may provide an alternative treatment for glaucoma that is independent of IOP lowering. However, it is possible that implementing a combination of treatments exploiting several mechanisms, such as IOP lowering, ECM remodeling, and neuroprotection, may provide the most effective treatment for patients. The resident cells of the ONH and ppScl that drive remodeling of these critical load-bearing connective tissues can potentially be recruited to improve the resilience of the optic nerve to glaucomatous injury. It is critical that additional research is conducted to clarify how the material properties in the LC and scleral should be adjusted to a beneficial effect and to elucidate the mechanocellular pathways involved in age-related and glaucomatous remodeling.

## Figures and Tables

**Figure 1 ijms-23-08068-f001:**
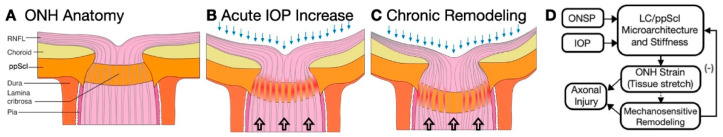
Mechanotransduction and optic nerve head remodeling. (**A**) Healthy optic nerve head (ONH) anatomy detailing key regions. (**B**,**C**) Increased intraocular pressure (IOP, blue arrows) is counterbalanced by optic nerve sheath pressure (ONSP, open arrows) resulting in tissue strain in the optic nerve head (ONH). This can damage axons directly (red) and activates cellular mechanotransduction that drives remodeling of the lamina cribrosa (LC) and peripapillary scleral (ppScl). (**C**) This remodeling alters the material properties and tissue architecture that modulates the stain that drives further remodeling. (**D**) This creates a negative feedback loop (−) that increases the vulnerability of the RGC axons to further glaucomatous injury. Deformation of any mechanical structure under load (strain) is determined by the loading forces (stress) along with its architecture and material properties.

**Figure 2 ijms-23-08068-f002:**
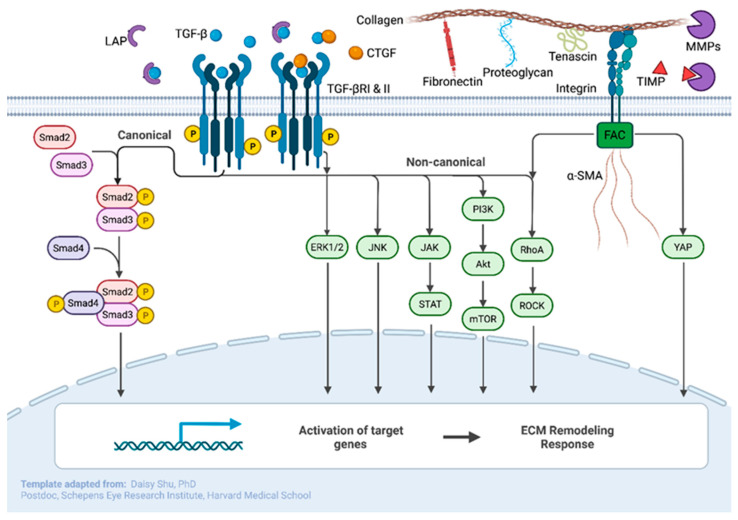
Canonical and non-canonical TGF-β pathways of importance and notable interactions with CTGF and integrin signaling. All pathways ultimately lead to alterations in ECM remodeling responses when activated. Adapted from “Canonical and Non-canonical TGFb Pathways in EMT”, by BioRender.com (2022). Available online: https://app.biorender.com/biorender-templates, accessed on 7 June 2022.

**Table 1 ijms-23-08068-t001:** Potential therapeutic targets to alter glaucomatous remodeling.

Mechanism of Action	Drug(s)	Impact on Optic Nerve Remodeling	Models Tested	References
Prostaglandin F receptor agonist	Bimatoprost, Latanoprost, Fluprostenol, Tafluprost, Travoprost	Upregulation of MMP-1, -3, -9	Mouse, Rat, Rabbit, Guinea Pig, Cat, Dog, Pig, Primate, Human	[131,132,133,134,135,136,137,138,139,140,141,142,143,144,145,146,147,148,149,150,151,152,153,154,155,156]
Hybrid prostaglandin F receptor agonist and nitric oxide donator	Latanoprostene bunod	Upregulation of MMPs and decrease cell contractility	Mouse, Rabbit, Dog, Primate, Human	[157,158,159]
β-adrenoceptor antagonist	Betaxolol, Timolol	Increased blood flow velocity	Mouse, Rat, Rabbit, Cat, Dog, Pig, Primate, Human	[160,161,162,163,164,165,166,167,168,169,170,171,172,173,174]
α_2_-adrenergic agonist	Apraclonidine, Brimonidine	Anti-apoptotic; RGC survival signal	Mouse, Rat, Guinea Pig, Rabbit, Cat, Dog, Pig, Primate, Human	[139,175,176,177,178,179,180,181,182,183,184,185,186,187,188]
Carbonic anhydrase inhibitor	Acetazolamide, Brinzolamide, Dorzolamide, Methazolamide	Increased blood flow and oxygen tension	Mouse, Rat, Guinea Pig, Rabbit, Dog, Pig, Primate, Human	[189,190,191,192,193,194,195,196,197,198,199,200,201,202,203,204,205,206,207,208,209,210]
ROCK Inhibitor	Fasudil, Netarsudil, Ripasudil	Inhibits contractility and migration of fibroblasts; inhibits production of ECM; inhibits cell death pathways	Mouse, Rat, Rabbit, Dog, Primate, Human	[211,212,213,214,215,216,217,218,219,220,221,222,223,224,225,226]
Inhibits secretion of TGF-β	Tranilast	Prevents TGF-β mediated fibrotic responses by nearby cells	Rabbit, Human culture	[227,228,229]
Inhibit transcription of TGF-β	ISTH0036, TbetaRII (RNAi)	Decreased levels of TGF-β expression	Mouse, Human Culture, Human	[230,231,232]
Direct immunosuppression of TGF-β	Lerdelimumab	Targeted inactivation of TGF-β to prevent receptor binding	Rabbit, Human	[233,234]
Inhibit TSP1 binding to LAP	LSKL	Inhibits TSP1 mediated activation of latent TGF-β	Mouse	[235]
Direct immunosuppression of CTGF	Pamrevlumab	Inhibits CTGF interaction with TGF-β	Human Culture	[37]
Reduce YAP and CTGF expression	Verteporfin (without light activation)	Reduces cell contractility via YAP; reduces CTGF interaction with TGF-β	Mouse, Human Culture, Human	[236,237,238]
Increased nitric oxide production	Atorvastatin, Lovastatin, Simvastatin	Inhibit RhoA/ROCK pathway and reduce levels of MMP-2 and -9, decrease cell contractility	Mouse, Rat, Rabbit, Dog, Pig, Human Culture	[65,239,240,241,242,243,244,245,246,247,248,249]
Angiotensin 1 receptor (AT1R) inhibitor	Losartan	Inhibits Smad2 phosphorylation	Mice, Rat, Rabbit, Human	[130,250,251,252]
Glycosaminoglycan degrading enzyme	Chondroitinase ABC	Weakens ECM (reduces stiffness)	Rat, Pig, Primate, Human Culture	[125,128,253,254]
Inhibit myosin light chain phosphorylation	Src-family tyrosine kinase (SFK) inhibitors (PP2)	Alters cell adhesion, reduces cell contractility, and permeability of cell layers	Rabbit, Human Culture	[108,255]

## Data Availability

Not applicable.

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
