# Peer review of "Remodeling of the Lamina Cribrosa: Mechanisms and Potential Therapeutic Approaches for Glaucoma"

_ijms, 2022, doi:10.3390/ijms23158068_

Round 1
Reviewer 1 Report
Dear Authors,
Pathophysiology of glaucoma remains still unknown. It is well known that lamina cribrosa is crutial and initial site for glaucomatous optic nerve degeneration. The manuscript presents elements of remodeling of lamina cribrosa during the corse of glaucoma in section: Remodeling response in aging and glaucoma, where role of extracellular matrix is mentioned, lamina cribrsa cells, gliall cells and fibroblasts, followed by discussion, where correlation with therapetic approaches is considered (table).
In my opinion the construction of the paper is correct and discusse in accordance with state of knowledge, however some potential elements are missing: the role of HIF-1 pathway, neuroinlammation, calcium and calcineurin-NFAT signaling pathway, transient receptor potential canonical channels.
Therefore I recommend acseptation for publication after minor revision.
Author Response
We appreciate both reviewers’ comments and have addressed these comments below. Overall, the focus of the review was on remodeling of the LC as a potential treatment. This is not to say that is the only non-IOP mechanism, just that this mechanism is the focus of the review. The following text was added to the introduction (lines 61-63) to clarify this: “While therapeutics aims at altering ECM remodeling are a promising potential treatment for glaucoma, it is not the only mechanism that can be exploited clinically.”
With regards to the specific comments, they are addressed below:
Review 1:
We added in Description of the other pathways mentioned by reviewer 1.
“the role of HIF-1 pathway”
Lines 230-241: “Astrocytic responses are not limited to mechanoreceptors as hypoxia can also trigger responses. Hypoxia-inducible factor-1α (HIF-1α) is a transcriptional factor that is upregulated in response to hypoxic conditions and plays a role in cellular metabolism, proliferation, and angiogenesis (Lee et al. 2020). The link between the glaucomatous ONH and HIF-1α was first noted by examining human eye post-mortem, but it has been noted in glaucomatous dogs as well (Tezel and Wax 2004; Savagian et al. 2008). These findings have been replicated in induced rodent models of glaucoma, and the evidence indicates that HIF-1α activation is localized to astrocytes of the retina and ONH (Ergorul et al. 2010; Chidlow et al. 2017). There is currently no explanation for why HIF-1α responses are localized to astrocytes and not microglia or RGCs in these models. However, it does indicate that global hypoxic conditions of the ONH, at least on these early timescales, cannot explain RGC dysfunction due to ischemia.”
“neuroinflammation”
While we agree that neuroinflammation is relevant to the glaucomatous ONH, we believe that mechanisms of neuroinflammation are more pertinent to RGC/axon health and processes of apoptosis/Wallerian degeneration and are outside of the scope of the ECM remodeling capabilities of glia. The perspective has also been recently reviewed and so we have cited those reviews as reference:
Lines 273-277: “The reactivity of both astrocytes and microglia contribute to neuroinflammatory conditions of the ONH that may negatively impact the surrounding milieu. This perspective of GON is complex and has generated a rapidly emerging line of work which has recently been adequately reviewed (Soto and Howell 2014; Mac Nair and Nickells 2015; Williams et al. 2017; Rolle et al. 2020; Tezel 2022).”
“calcium and calcineurin-NFAT signaling pathway”
Lines 164-176: “TGF-β, as well as mechanical and oxidative stress, can also influence aspects of ECM regulation in LC cells through calcium dependent pathways. For instance, LC cells cultured from glaucomatous human eyes demonstrated dysregulation of calcium, such as high intracellular levels in response to previously mentioned stimuli and a reduced ability to sequester free cytoplasmic calcium (Irnaten et al. 2013; Irnaten et al. 2018). Increased levels of cellular calcium can have a variety of effects on signaling pathways due to the dynamic nature of calcium as a second messenger. Of those pathways, the activation of nuclear factor of activated T-cells (NFAT) may be most relevant to LC cells. In short, calcineurin can bind calcium to calmodulin, which can dephosphorylate NFAT. NFAT then complexes with transcription factors to influence transcription of genes including those that modulate the ECM (Tidu et al. 2021). While inhibition of this pathway may aid in the treatment of glaucoma, it is not fully understood what precedes the loss of calcium regulation in glaucomatous LC cells.”
“transient receptor potential canonical channels”
Lines: 176- 191: “However, one potential preceding factor may be the presence of transient receptor potential canonical (TRPC) channels, a class of voltage-independent channels that preferentially respond to calcium ions and are not necessarily dependent on stimuli such as TGF-β or oxidative and mechanical stresses (Sharma and Hopkins 2019; Asghar and Törnquist 2020;). Interestingly, isoforms of TRPCs, such as 1 and 6, are significantly overexpressed in glaucomatous LC cells cultured from human ONHs (Irnaten et al. 2020). The increased presence of these channels may disturb homeostatic levels of calcium in LC cells, thereby inducing dysregulation. Also, these channels modulate transcription of ECM components such as TGF-β, α-SMA, collagen, and MMPs likely through the NFAT pathway described previously (Irnaten et al. 2020; Asghar and Törnquist 2020). Furthermore, TRPC-1 and -6, at least in cancerous cells of the central nervous system, also regulate cell migration (Chigurupati et al. 2010; Asghar and Bomben et al. 2011; Törnquist 2020). While these mechanisms of migration have not directly been shown in LC cells, it is possible that glaucomatous LC cells overexpressing these TRPCs may initiate signaling cascades that increase the degree to which LC cells migrate within the LC, potentially contributing to posterior LC migration.”
And
Lines: 224-229: “Astrocytes also detect mechanical strain with TRPCs which may provide early responses to initial IOP increases. For example, in an induced mouse model, reactive astrocytes respond within one hour of an IOP increase, likely mediated by TRPC isoforms sensitive to stretch (Choi et al. 2015). As previously mentioned, this TRPC-NFAT pathway can induce transcriptional changes related to the ECM and can influence cellular migration”.
Reviewer 2 Report
The work focused on the biochemical aspects of the LC in glaucoma. However, these changes may be secondary. This theory cannot explain the primary damage to retinal ganglion cell axons first retrolaminarly and then prelaminarly. This fact has also been demonstrated experimentally in cadaveric glaucoma eyes. Target excavation may arise not only by high IOP but by loss of nerve fibers retrolaminarly.
Otherwise, in glaucoma there is primary damage to the ganglion cells of the retina and only secondarily to their axons.
I would appreciate at least a minor mention of mechanisms of RNFL damage in glaucoma other than dogmatically focusing on the LC level.
Author Response
We appreciate both reviewers’ comments and have addressed these comments below. Overall, the focus of the review was on remodeling of the LC as a potential treatment. This is not to say that is the only non-IOP mechanism, just that this mechanism is the focus of the review. The following text was added to the introduction (lines 61-63) to clarify this: “While therapeutics aims at altering ECM remodeling are a promising potential treatment for glaucoma, it is not the only mechanism that can be exploited clinically.”
With regards to the specific comments, they are addressed below:
Review 2:
To clarify, we are not asserting that ONH remodeling is the only mechanism responsible for glaucoma pathogenesis, only that it plays a significant and interactive role in overall glaucoma pathogenesis. This review was to focus on the possibly mechanisms involved in the remodeling of the LC and ppScl that could be use as potential target to treat glaucoma. This does not exclude the multiple other mechanism that are undoubtedly playing a role. We have clarified this in the statements below.
With regards to the specific comments:
"This theory cannot explain the primary damage to retinal ganglion cell axons first retrolaminarly and then prelaminarly.
While we agree that changes in the retrolaminar or prelaminar ONH or even peripheral retina can occur and potentially be causal in glaucoma, we respectfully disagree with reviewer on this assertion. Changes in the morphology and material properties of the load bearing structures of the optic nerve could also impact pre- and retro-laminar tissue strain and perfusion along with surrounding peripapillary choroidal perfusion. Indeed, mechanically, one would expect the largest sheer strains at the interfaces where axonal bundles enter the firmly supported LC especially in these regions. Altering the mechanical properties of the principle mechanical structural support of the ONH will impact strain across its entirety. In short, all these tissues are all connected and thus move together when stretched.
This fact has also been demonstrated experimentally in cadaveric glaucoma eyes.
Remodeling driven changes in the mechanical behavior of the ONH’s load bearing tissues can still impact the pre and retrolaminar region. The finding the reviewer mentions does not exclude an impact from remodeling of the load bearing connective tissue of the LC, which remains a well-documented pathologic feature of glaucoma not seen to the same degree in other optic neuropathies. Lastly, it is difficult to determine causality from cadaveric studies, given death is the cross-sectional event that it is. We are not aware of human cadaveric study that definitively demonstrates, or suggests, that remodeling of the LC could not impact a potential site of injury beginning in the pre- or retrolaminar space.
Target excavation may arise not only by high IOP but by loss of nerve fibers retrolaminarly.
Certainly, loss of RNFL can cause enlargement of the cup, as seen in compressive and other optic neuropathies. However, other neuropathies do not have the concomitant remodeling of the LC and ppScl to the extent seen in the pathology associated with most cases of glaucoma. The article is focused on intervention targeted to the mechanism of ECM remodeling. This does not exclude the many other pathologies beyond the scope of this focused review.
Otherwise, in glaucoma there is primary damage to the ganglion cells of the retina and only secondarily to their axons.
Our review focused on remodeling of the ECM as a particular target for glaucoma. I do not think that there is a literature that enable the conclusion that RGC soma are damaged primarily in glaucoma. It may of course happen, but I do not agree that there is strong evidence for this statement. I am happy for the reviewer to provide such evidence.
I would appreciate at least a minor mention of mechanisms of RNFL damage in glaucoma other than dogmatically focusing on the LC level."
In our response to review 1, we have added a statement emphasizing that this review is focused on alterations in ECM remodeling as a potential treatment for glaucoma. This does not mean we feel this is the only treatment avenue, but is one mechanism, of many, that can be exploited.
Lines 61-63: “While therapeutics aims at altering ECM remodeling are a promising potential treatment for glaucoma, it is not the only mechanism that can be exploited clinically.”